# Evaluation of T Cell Response to SARS-CoV-2 in Kidney Transplant Recipients Receiving Monoclonal Antibody Prophylaxis and the Utility of a Bivalent mRNA Vaccine Booster Dose

**DOI:** 10.3390/microorganisms12040722

**Published:** 2024-04-02

**Authors:** Dominique Bertrand, Charlotte Laurent, Mathilde Lemoine, Ludivine Lebourg, Mélanie Hanoy, Frank Le Roy, Dorian Nezam, Diana Pruteanu, Steven Grange, Tristan De Nattes, Véronique Lemée, Dominique Guerrot, Sophie Candon

**Affiliations:** 1Department of Nephrology, Transplantation and Hemodialysis, Rouen University Hospital, 76000 Rouen, France; charlotte.laurent@chu-rouen.fr (C.L.); mathilde.lemoine@chu-rouen.fr (M.L.); ludivine.lebourg@chu-rouen.fr (L.L.); melanie.hanoy@chu-rouen.fr (M.H.); frank.le-roy@chu-rouen.fr (F.L.R.); dorian.nezam@chu-rouen.fr (D.N.); diana.pruteanu@chu-rouen.fr (D.P.); steven.grange@chu-rouen.fr (S.G.); tristan.de-nattes@chu-rouen.fr (T.D.N.); dominique.guerrot@chu-rouen.fr (D.G.); 2INSERM U1234, University of Rouen Normandy, 76000 Rouen, France; sophie.candon@chu-rouen.fr; 3Department of Virology, Rouen University Hospital, 76000 Rouen, France; veronique.lemee@chu-rouen.fr; 4INSERM U1096, University of Rouen Normandy, 76000 Rouen, France; 5Department of Immunology and Biotherapies, Rouen University Hospital, 76000 Rouen, France

**Keywords:** belatacept, kidney transplantation, COVID-19, cellular immune response, ELISPOT

## Abstract

Monoclonal antibodies have been administered to kidney transplant recipients (KTRs) with a poor or non-responder status to SARS-CoV-2 vaccination. The cellular response to SARS-CoV-2 has been poorly studied in this context. We assessed the T cell response to SARS-CoV-2 in 97 patients on the day of the injection of tixagevimab/cilgavimab using an IFNγ enzyme-linked immunospot assay (ELISPOT). Among the 97 patients, 34 (35%) developed COVID-19 before the injection. Twenty-nine (85.3%) had an ELISPOT compatible with a SARS-CoV-2 infection. There was no difference between KTRs under belatacept or tacrolimus treatment. Sixty-three patients (64.9%) had no known COVID-19 prior to the ELISPOT, but nine (14.3%) had a positive ELISPOT. In 21 KTRs with a positive ELISPOT who received a booster dose of a bivalent mRNA vaccine, median antibody titers and spike-reactive T cells increased significantly in patients under tacrolimus but not belatacept. Our study emphasizes the potential usefulness of the exploration of immune cellular response to SARS-CoV-2 by ELISPOT. In KTRs with a positive ELISPOT and under CNI therapy, a booster dose of mRNA vaccine seems effective in inducing an immune response to SARS-CoV-2.

## 1. Introduction

Severe acute respiratory syndrome coronavirus 2 (SARS-CoV-2) infection caused a global pandemic that affected France in March 2020. Over 3 years, more than 160,000 people died [1], particularly immunocompromised persons like kidney transplant recipients (KTRs), for whom the mortality rate was close to 22% at the start of the pandemic [2]. One response to this pandemic was vaccination, which was rapidly recommended through international guidelines [3,4]. The Omicron variant was identified in November 2021 in Botswana, South Africa and quickly spread worldwide to become the predominant variant. Due to its immune escape profile, vaccination results had reduced neutralizing activity against Omicron, compared with the ancestral strain [5].

COVID-19 remains a major problem in renal transplantation because of the morbidity and mortality induced in these patients [6]. Indeed, despite a reinforced vaccination schedule with three or four doses of mRNA vaccines [7], a significant proportion of patients remain poorly or not protected against SARS-CoV-2, with a high risk of breakthrough severe infection [8]. In this context, monoclonal antibodies directed against the virus have been developed and used in kidney transplant recipients (KTR), either as curative [9] or prophylactic treatment [10], to enhance immunity against SARS-CoV-2 in immunocompromised patients. For example, tixagevimab–cilgavimab is a combination of two fully human, SARS-CoV-2-neutralizing monoclonal antibodies, which are derived from antibodies isolated from B cells obtained by persons infected with SARS-CoV-2. The PROVENT study assessed tixagevimab–cilgavimab for pre-exposure prophylaxis against symptomatic COVID-19. Relative risk reduction for symptomatic COVID-19 was 76.7% in the tixagevimab–cilgavimab group. Efficacy is estimated to last at least 6 months [10]. Tixagevimab–cilgavimab can also be used as an early treatment in high-risk patients developing moderate-to-severe COVID-19 [11]. The results of the use of these antibodies are encouraging in KTRs [12,13], although there are arguments for an escape of sensitivity of the new variants and, in particular, of Omicron [14]. In France, the prophylactic use of monoclonal antibodies (casirivimab–imdevimab, from 12 September 2021; tixagevimab/cilgavimab, from December 2021) was recommended in solid organ transplantation patients with a complete vaccine scheme and no or weak humoral response (<264 binding antibody units [BAU]/mL [15]). During the COVID-19 pandemic, the FDA granted EUAs for monoclonal antibody therapies, such as tixagevimab/cilgavimab and sotrovimab. However, both therapies lost their authorization due to their inefficacy against newer Omicron subvariants.

Innate and adaptive (both T and B cells) immune populations work synergistically to prevent, limit, and clear SARS-CoV-2 infection [16]. Early studies suggest that CD8^+^ T cells are critical for mediating the anti-viral response in acute infection, while B and CD4^+^ T cells stand out for their roles in the prevention of viral infection and ultimate viral clearance [16]. Although humoral immune response has been extensively explored and reported in immunocompromised subjects [17], data on the cellular response after COVID-19 and the SARS-CoV-2 vaccine are more scarce, even more so in KTRs under monoclonal antibody prophylaxis.

We report in our study the evaluation of anti-SARS-CoV-2-specific T cell immunity in KTRs receiving monoclonal antibody prophylaxis. We demonstrate that the ELISPOT is a useful tool in this context to recognize undocumented COVID-19 infection and screen for patients that may benefit from a booster vaccine dose. We also depict the immune response to a booster dose of a bivalent mRNA vaccine in 21 KTRs with a positive ELISPOT.

## 2. Materials and Methods

### 2.1. Patients

Between 23 December 2021 and 7 February 2022, 412 renal transplant patients who had weak or no response to the anti-SARS-CoV-2 vaccine received an intramuscular injection of 150 mg of tixagevimab + 150 mg of cilgavimab [12], followed by a 2nd injection in May 2022. The present study included the first 97 KTRs, in which the cellular immune response to SARS-CoV-2 was assessed on the day of the 2nd injection. All patients were instructed to systematically report potential symptoms of COVID-19 and/or the positivity of a nasopharyngeal swab for SARS-CoV-2. A follow-up was set for 1 October 2022. Among the 97 KTRs, 21 KTRs showing SARS-CoV-2-reactive T cells suggestive of a recent SARS-CoV-2 infection received a booster dose of an mRNA vaccine (bivalent Spikevax, Moderna), and both humoral and cellular responses were assessed on the day of the injection and 3 weeks later.

According to French law (Loi Jardé), because this was an anonymous retrospective study, institutional review board approval was not required. The clinical and research activities being reported are consistent with the principles of the Declaration of Istanbul, as outlined in the ‘Declaration of Istanbul on Organ Trafficking and Transplant Tourism’.

### 2.2. Interferon-γ Enzyme-Linked Immunospot Assay

Peripheral blood mononuclear cells were isolated by density gradient centrifugation of blood samples and used immediately. Peripheral blood mononuclear cells (in concentrations adjusted to 2 × 10^5^ CD3^+^ T cells per well) were plated in ELISPOT 96-well plates in the presence of overlapping 15-mer peptide pools spanning the sequence of SARS-CoV-2, with the time-structural and non-structural accessory proteins S (pool S1 spanning the N-terminal part of the protein, including the S1 subunit, and pool S2 spanning the C-terminal part), N, M, E, NS3A, and NS7A (JPT, Strassberg, Germany). Negative and positive control stimulations, medium only, and CEFX peptide pool (JPT, Strassberg, Germany), respectively, were included in the assay. After overnight culture, the cells were washed, and the captured IFN-γ was revealed using a colorimetric assay (UCytech, Utrecht, The Netherlands). Spots were counted with an automated ELISPOT reader (AID, Strassberg, Germany). For each stimulation condition, the average spot number observed in wells without the antigen was subtracted. Results were expressed as the SFC (spot-forming cells) per 106 CD3^+^ T cells. For each assay, a specific response was considered positive if the SFC number was superior to 3 SDs of the mean of the spot numbers observed in wells without antigens (negative control) (25 SFC/10^6^ CD3^+^ T cells) [18].

ELISPOT was considered suggestive of SARS-CoV-2 infection (S+/N+) if there was a specific response to non-spike antigens with a response against S1 and/or S2 (S1 + S2 = S). It was considered a vaccine response (S+/N−) if there was a specific response only against S1 and/or S2. ELISPOT was considered negative if the responses for S and N were negative (S−/N−).

### 2.3. Anti–SARS-CoV-2 Antibody Response

Anti-spike receptor-binding domain IgG antibodies were measured using the ARCHITECT IgG II Quant test (Abbott, Abbott Park, IL, USA) (positivity threshold: 8.5 binding antibody units per mL [BAU/mL]) [19].

### 2.4. Statistical Methods

Quantitative data were presented as the mean (SD) or median (interquartile range: IQR) when data were not normally distributed. Qualitative data were presented as percentages. The nonparametric Mann–Whitney test (quantitative data) and the chi-square test (qualitative data) were used to compare characteristics between the 2 groups. To demonstrate a correlation between time and cellular immune response, we performed a Pearson correlation test, with a correlation coefficient (r) r < 0.25 indicating low correlation, 0.25 < r < 0.5 indicating moderate correlation, 0.5 < r < 0.75 indicating strong correlation, and r > 0.75 indicating excellent correlation. All analyses were performed using StatView version 5.0 (SAS Institute, Cary, NC, USA), and graphs were generated using the GraphPad Prism version 8.0 software (GraphPad Software, San Diego, CA, USA).

## 3. Results

### 3.1. General Characteristics of Patients and COVID-19

General characteristics of the 97 KTRs included are summarized in Table 1. Fifty-eight KTRs were under the tacrolimus regimen on the day of the evaluation (tacrolimus group: 59.8%), whereas thirty-nine were under the belatacept regimen (belatacept group: 40.2%). None of them had a history of SARS-CoV-2 infection before the first injection of tixagevimab/cilgavimab.

Between 23 December 2021 and 17 May 2022, 34 (35%), SARS-CoV-2 infections (Omicron variant) with a median delay of 102.5 days (IQR: 43.5, 139.7) were developed before the second injection: 21 symptomatic infections (21.6%), including 7 hospitalizations (7.2%), 2 of which were in the ICU (2.1). In the belatacept group, 17/39 (43.6%), and 17/58 (29.3%) in the tacrolimus group, had a SARS-CoV-2 infection. Patients hospitalized for SARS-CoV-2 infection were significatively more frequent in the belatacept group compared to the tacrolimus group (15.4% vs. 1.7%, *p* = 0.01).

### 3.2. Cellular Immune Response to SARS-CoV-2 in KTR with History of Omicron Infection

Among the 34 KTRs with a previous Omicron SARS-CoV-2 infection, 29 (85.3%) had a positive ELISPOT with a specific response against the spike (S+) and non-spike antigens (N+), compatible with the history of COVID-19. The median numbers of spike-reactive T cells and N-reactive T cells were 495 (IQR: 185–800) and 240 (IQR: 115–417.5) SFC/10^6^ CD3^+^ T cells, respectively. Three KTRs (8.8%) had a positive ELISPOT with spike-reactive T cells but without non-spike-reactive T cells. One KTR had a positive ELISPOT with non-spike-reactive T cells but without spike-reactive T cells (2.9%), and one KTR (2.9%) had a negative ELISPOT.

The median number of reactive T cells (S1, S2, N, M, and NS3A) was significatively higher in KTRs with a history of SARS-CoV-2 infection, compared to COVID-19-negative KTRs (Figure 1a). The median number of spike-reactive T cells (S1 + S2) was significatively higher in KTRs with a history of symptomatic SARS-CoV-2 infection, compared to asymptomatic infection (Figure 1b): 590 (IQR: 247.5–925) vs. 145 (IQR: 60–507.5) SFC/10^6^ CD3^+^ T cells (*p* = 0.01). The median number of N-reactive T cells was not significatively higher: 245 (IQR: 100–452.5) vs. 115 (IQR: 40–272.5) SFC/10^6^ CD3^+^ T cells (*p* = 0.11). Among KTRs with previous symptomatic infections, numbers of reactive T cells were not statistically different between patients who were hospitalized and those who were not.

The number of spike- and N-reactive T cells did not correlate with the time since the last documented SARS-CoV-2 infection (Figure 2).

In the belatacept group, 16/17 KTR (94.1%) had a positive ELISPOT with a specific response against the spike (S+) and non-spike antigens (N+), while there were 13/17 (76.5) in the tacrolimus group (*p* = NS). Median numbers of spike-reactive T cells were not significantly different between the tacrolimus and the belatacept groups (Figure 3a): 470 (IQR: 97.5–767.5) vs. 430 (IQR: 170–887.5) SFC/10^6^ CD3^+^ T cells (*p* = 0.46). Median numbers of N-reactive T cells were also similar in both groups: 120 (IQR: 55–375) vs. 220 (IQR: 67.5–407.5) SFC/10^6^ CD3^+^ T cells (*p* = 0.55).

### 3.3. Cellular Immune Response to SARS-CoV-2 in KTRs with no History of Omicron Infection

Sixty-three patients (64.9%) had no known SARS-CoV-2 infection prior to the assessment of anti-SARS-CoV-2 T cell immunity. Among them, nine (14.3%) had a positive ELISPOT with a specific response against the spike (S+) and non-spike antigens (N+), compatible with a history of COVID-19. Median numbers of spike-reactive T cells and N-reactive T cells were 270 (IQR: 137.5–432.5) and 195 (IQR: 67.5–302.5) SFC/10^6^ CD3^+^ T cells, respectively. Thirteen KTRs (21.1%) had a positive ELISPOT with spike-reactive T cells but without non-spike-reactive T cells. One KTR had a positive ELISPOT with non-spike-reactive T cells but without spike-reactive T cells (1.6%), and forty KTRs (63.5%) had a negative ELISPOT.

Table 2 summarizes the results of the ELISPOT, according to the belatacept or tacrolimus group, in KTRs with or without a history of COVID-19.

### 3.4. Evolution after the Second Injection of Tixagevimab/Cilgavimab

After the second injection of tixagevimab/cilgavimab and a median follow up of 122 days (IQR: 114–130), five patients developed a SARS-CoV-2 infection; four were symptomatic, and one KTR was hospitalized. Among the five patients, four had a negative ELISPOT, and one had a positive ELISPOT with spike-reactive T cells but without non-spike-reactive T cells. No case of reinfection was reported.

### 3.5. Anti-SARS-CoV-2-Specific Immune Response after a Booster Dose of a Bivalent mRNA Vaccine (Bivalent SPIKEVAX, Moderna) (Figure 4)

Among the 97 KTRs, 21 KTRs received a booster dose of the mRNA vaccine (bivalent Spikevax, Moderna). Twelve KTRs were under tacrolimus therapy, and nine were under belatacept therapy at the time of the injection.

Regarding humoral response, median antibody titers increased from 1431 BAU/mL (IQR, 856–2240) to 1836 BAU/mL (IQR, 1033–3980, *p* = 0.04). This increase was statistically significant in the tacrolimus group, which was from 1384 BAU/mL (IQR, 805–2319) to 2964 BAU/mL (IQR, 1510–4231, *p* = 0.01), but not in the belatacept group, which was from 1431 BAU/mL (IQR, 856–2140) to 1281 BAU/mL (IQR, 759–2050, *p* = 0.9).

Regarding cellular response, the median number of spike-reactive T cells increased from 245 SFCs/10^6^ CD3^+^ T cells (IQR, 80–372) to 718 SFCs/10^6^ CD3^+^ T cells (IQR: 302–1242, *p* = 0.01). This increase was statistically significant in the tacrolimus group, which was from 190 SFCs/106 CD3^+^ T cells (IQR: 77–345) to 955 SFCs/106 CD3^+^ T cells (IQR: 283–1231, *p* = 0.008), but not in the belatacept group, which was from 255 BAU/mL (IQR, 100–547) to 515 BAU/mL (IQR, 377–2050, *p* = 0.21).

## 4. Discussion

We report here the first study assessing the T cell response to SARS-CoV-2 in a cohort of KTRs with a poor or non-responder status to SARS-CoV-2 vaccination under tixagevimab/cilgavimab prophylaxis. We confirmed that despite a poor vaccinal response, the majority of KTRs are able to develop a cellular response to SARS-CoV-2 when infected, regardless of the intensity and severity of the disease. This response was stronger in symptomatic than asymptomatic KTRs. This point has already been reported in literature, but it was before the era of vaccination [20]. Data suggest that the SOT population achieved comparable functional immune responses, although with some initial delay, to those of the general population after moderate/severe COVID-19 [21] and is capable of maintaining long-lasting peripheral immune memory after COVID-19 infection, mainly determined by the degree of infection severity [22]. One striking feature in the present study is that KTRs under belatacept therapy, known to be poor responders to vaccines [23], were able to develop a strong cellular-specific immune response when infected, comparable to that observed in patients under tacrolimus therapy. This observation suggests that adjuvant signals provided by the natural virus are able to bypass the co-stimulation blockade in these patients, allowing for the priming of a specific T cell response. However, this response might not be optimal, since hospitalization for COVID-19 was significatively more frequent in the belatacept group, compared to the tacrolimus group (15.4% vs. 1.7%, *p* = 0.01). Nevertheless, confounding factors might be the cause of such a difference between the two groups.

The SARS-CoV-2 ELISPOT seems to be a good tool for detecting KTRs with asymptomatic and unproven COVID-19. Among the cohort of KTRs without a history of prior SARS-CoV-2 infection, we reported that 15% of them had an undetected infection. Those KTRs developed a significant cellular response comparable to that of KTRs with proven asymptomatic infection. The assessment of T cell immunity by ELISPOT or other interferon-γ release assays for T cell-mediated immunity (IGRA) can be easily implemented in a routine setting. At present, two such tests based on interferon-gamma release are available—Quan-T-Cell SARS-CoV-2 by EUROIMMUN and T-SPOT.COVID by Oxford Immunotec; the first head-to-head comparison of these two tests evaluating T cell-mediated immunity against SARS-CoV-2 was recently published and showed comparable results between the two tests [24]. For example, assessing specific T cell-mediated immunity against cytomegalovirus (CMV) holds the potential to enhance personalized strategies aimed at preventing and treating CMV in organ transplantation. Robust evidence has showed a close association between CMV cell-mediated immunity, especially CMV-specific CD4^+^ and CD8^+^ T lymphocytes, and the risk of developing a CMV infection in different transplant settings [25,26]. Current evidence indicates that cell-mediated immune assays are helpful in identifying patients at low risk for replication, for whom preventive measures against CMV can be safely withheld [27]. These tests are the only way to detect unnoticed infection because serology cannot help in this context, as the detected anti-spike antibodies are infused monoclonal antibodies. Furthermore, evaluation of T cell immunity by ELISPOT may also help to differentiate SARS-CoV-2 infection from COVID-19 post-vaccinal response [28], particularly in KTR non-responders to vaccine under monoclonal antibody prophylaxis. Among the entire cohort, 16/97 (16%) had a positive ELISPOT with spike-reactive T cells but without non-spike-reactive T cells, suggestive of a cellular response induced by the vaccine.

We know that the SARS-CoV-2 vaccination is particularly effective in patients with a past history of COVID-19 [19], even in the KTR population [29]. We show here that in KTRs, non-responders to the vaccine but with a history of a previous SARS-CoV-2 infection and/or a positive ELISPOT (S+/N+), a booster dose of the mRNA vaccine increased both humoral and cellular-specific immune responses. Unfortunately, this increase was significant only in tacrolimus-treated patients but not in patients under belatacept. Chavarot et al. already reported that patients under belatacept [23] therapy do not respond to the vaccine and are particularly exposed to opportunistic infections [30]. Nevertheless, in non-belatacept-treated patients, a booster dose of the mRNA vaccine after the infection could be proposed. ELISPOT could be particularly useful in patients with an unnoticed infection, as they could benefit from a new vaccine injection and strengthen their immunity against SARS-CoV-2. We would like to emphasize this point in the context of the Omicron SARS-CoV-2 era, in which new variants (BQ.1.1 and XBB) have immune-evasion capabilities that are greater than those of earlier omicron variants [31] and are not more sensitive to monoclonal antibodies such as tixagevimab/cilgavimab. A booster dose of vaccine, even in initial non-responders, may be beneficial in preventing severe forms of future waves of COVID-19. This strategy has to be confirmed in a prospective cohort of patients to ensure that this booster dose is readily associated with the induction of an immune response in patients not responding after 3, 4, or 5 doses of vaccines. In patients with a positive ELISPOT S+/N−, the same strategy could be proposed.

Nevertheless, a large proportion of KTRs (41/97) had a negative ELISPOT (S−/N−) and no immune response to the vaccine. These KTRs probably have a high risk of a severe form of COVID-19. Prophylaxis with monoclonal antibodies was helpful in this context, leading to less infection, symptomatic or not, less hospitalization, less requirement of intensive care, and lower mortality [12]. In this era of immune-evasion capabilities, the continued evolution of Omicron variants reinforces the need for new therapeutic monoclonal antibodies for COVID-19. In the wake of this, antivirals such as nirmetralvir/ritonavir [32] could be proposed, but reports of their use in KTRs are very scarce [33], due to their important interactions with CNI and their limitation of use for eGFRs below 30 mL/min. In the recently published study by Yang et al., 40 KTRs with severe renal impairment (estimated glomerular filtration rate < 30 mL/min) were included and treated with nirmatrelvir–ritonavir; none of the 32 moderate patients experienced disease progression. However, among the eight patients with critical COVID-19, unfortunately, two of them died. During the medication period, four patients experienced a total of six adverse events associated with nirmatrelvir–ritonavir. These findings supported the use of nirmatrelvir–ritonavir therapy for the treatment of COVID-19 in transplant patients with severe renal insufficiency. The modified dose of nirmatrelvir–ritonavir was well-tolerated [34].

We are well aware of the limitations of our study. The major limitation is the size of the reported cohort, which, overall, is very small in the assessment of the immune response induced by the booster dose of the mRNA vaccine. Nevertheless, we believe that the SARS-CoV-2 ELISPOT is a useful assay to determine the specific cellular immune response in KTRs in order to determine which KTR may benefit from an extra dose of vaccine.

In summary, our study emphasizes the potential usefulness of the assessment of T cell immunity to SARS-CoV-2 by ELISPOT in a cohort of KTRs at high risk of COVID-19 who were treated with tixagevimab/cilgavimab. First, the assay confirmed that despite the poor response to the vaccine, the majority of KTRs are able to develop a cellular response to SARS-CoV-2 infection. Secondly, it might be a good tool to differentiate asymptomatic, unproven infection from an isolated cellular response to the vaccine. This could be important for the prevention of future COVID-19 and for detecting KTRs likely to respond to a booster dose of the mRNA vaccine. Lastly, we demonstrated again that some of our patients are not protected and are still at high risk of a severe form of COVID-19, especially those under belatacept therapy, in this era of immune-escape Omicron variants.

## Figures and Tables

**Figure 1 microorganisms-12-00722-f001:**
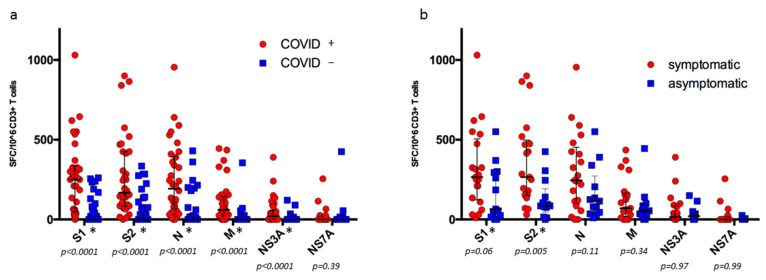
SARS-CoV-2-reactive IFN*γ*-producing T cells according to the (**a**) history of Omicron infection (all cohorts, n = 97) and (**b**) the symptomatic nature of the infection in KTRs with a history of Omicron infection (n = 34). * statistically different between the 2 groups.

**Figure 2 microorganisms-12-00722-f002:**
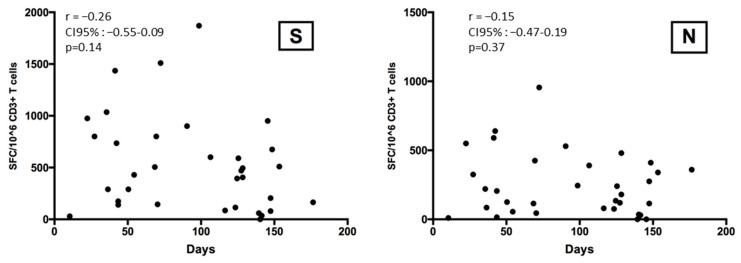
Spike- and N-reactive T cells in patients according to the time (days) since Omicron infection.

**Figure 3 microorganisms-12-00722-f003:**
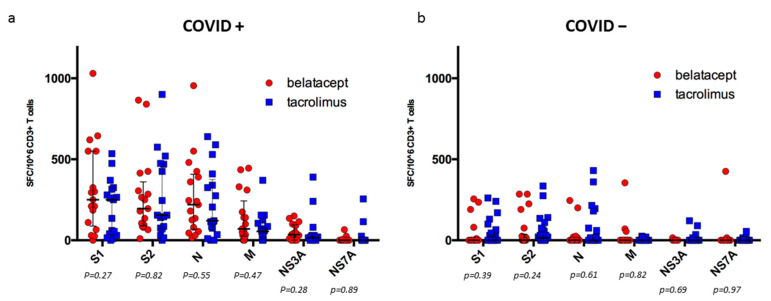
SARS-CoV-2-reactive IFN*γ*-producing T cells in (**a**) KTRs with history of Omicron infection and in (**b**) KTRs with no history of Omicron infection.

**Figure 4 microorganisms-12-00722-f004:**
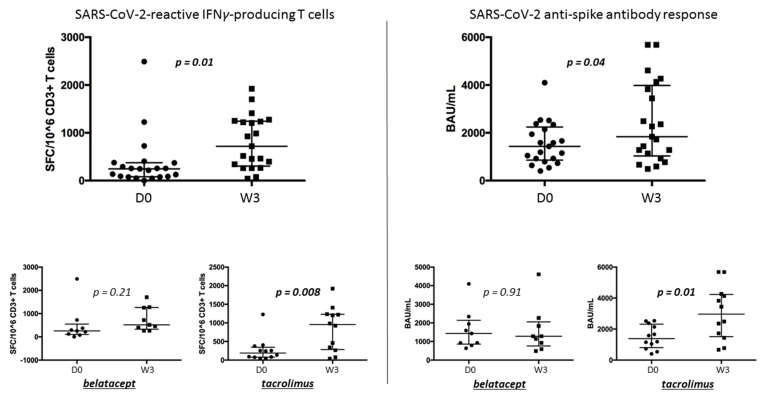
SARS-CoV-2–reactive IFN*γ*-producing T cells and SARS-CoV-2 anti-spike antibody response in KTRs before (D0) and 3 weeks (W3) after the mRNA bivalent vaccine booster dose.

**Table 1 microorganisms-12-00722-t001:** General characteristics of patients. M: male; F: female; eGFR: estimated glomerular filtration rate; DSA: donor-specific antibody; MMF: mycophenolate mofetil; AZA: azathioprine; ICU: intensive care unit. * Statistical comparison between tacrolimus group and belatacept Group.

	Total n = 97	Tacrolimus Group n = 58	Belatacept Group n = 39	*p* *
Age—years	63.1 ± 13.4	62.9 ± 14.5	63.3 ± 1.9	0.84
Sex M/F—n (%)	59 (60.8)/38 (39.2)	34 (58.6)/24 (41.4)	25 (64.1)/14 (35.9)	0.58
Time from transplantation—years	5.3 (2.7–9.6)	4.9 (2.4–9.1)	5.7 (2.9–11.0)	0.19
eGFR—mL/min/1.73 m^2^	44.9 ± 19.9	51.5 ± 20.1	35,2 ± 15.3	<0.0001
DSA n (%)	13 (13.4)	6 (10.3)	7 (17.9)	0.28
Immunosuppression—n (%)				
Tacrolimus	58 (59.8)	58 (100)	0 (0)	<0.0001
MMF	83 (85.5)	50 (86.2)	33 (84.6)	0.62
AZA	34 (12.4)	2 (3.5)	5 (12.8)	0.14
Mtor Inhibitors	2 (2)	1 (1.7)	1 (2.6)	0.46
Belatacept	39 (40.2)	0 (0)	39 (100)	<0.0001
Steroids	51 (52.5)	31 (53.4)	20 (51.3)	0.70
COVID-19 + n (%)	34 (35.1)	17 (29.3)	17 (43.6)	0.14
Symptomatic n (%)	21 (21.6)	11 (18.9)	10 (25.6)	0.43
Hospitalized n (%)	7 (7.2)	1 (1.7)	6 (15.4)	0.01
ICU requirement n (%)	2 (2.1)	0 (0)	2 (5.1)	0.08

**Table 2 microorganisms-12-00722-t002:** Results of the SARS-CoV-2 ELISPOT in (a) KTRs with a history of Omicron infection and (b) KTRs with no history of Omicron infection.

**(a)**				
	**Total** **n = 34**	**Tacrolimus Group** **n = 17**	**Belatacept Group** **n = 17**	** *p* **
S+/N+ n (%)	29 (85.3)	13 (76.4)	16 (94.1)	NS
S+/N− n (%)	3 (8.9)	2 (11.8)	1 (5.9)	NS
S−/N+ n (%)	1 (2.9)	1 (5.9)	0 (0)	NS
S−/N− n (%)	1 (2.9)	1 (5.9)	0 (0)	NS
**(b)**				
	**Total** **n = 63**	**Tacrolimus Group** **n = 41**	**Belatacept Group** **n = 22**	** *p* **
S+/N+ n (%)	9 (14.3)	6 (14.6)	3 (13.6)	NS
S+/N− n (%)	13 (20.6)	10 (24.4)	3 (13.6)	NS
S−/N+ n (%)	1 (1.6)	1 (2.4)	0 (0)	NS
S−/N− n (%)	40 (63.5)	24 (58.6)	16 (72.7)	NS

## Data Availability

All relevant data are within the paper.

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
