# Peer review of "Evaluation of T Cell Response to SARS-CoV-2 in Kidney Transplant Recipients Receiving Monoclonal Antibody Prophylaxis and the Utility of a Bivalent mRNA Vaccine Booster Dose"

_microorganisms, 2024, doi:10.3390/microorganisms12040722_

Round 1

Reviewer 1 Report

Comments and Suggestions for Authors

Academic degrees are not necessary in the authors section

"COVID-19" line 39 and 230

Line 94 put the meaning of SFC and put the appropriate scientific notation

Check the table headers as they are not visible.

How would the use of these monoclonal antibodies affect new variants of SARS-CoV-2, what would be the strategy of use? Perhaps that is why some of his patients did not present COVID-19.

All patients who developed COVID-19 survived, were there complications resulting from this?

Author Response

- Academic degrees are not necessary in the authors section

We modified this point in the revised manuscript

- "COVID-19" line 39 and 230

We modified this point in the revised manuscript

- Line 94 put the meaning of SFC and put the appropriate scientific notation

We add the meaning of SFC in the text: Spot Forming Cells.

- Check the table headers as they are not visible.

We modified this point in the revised manuscript

- How would the use of these monoclonal antibodies affect new variants of SARS-CoV-2, what would be the strategy of use? Perhaps that is why some of his patients did not present COVID-19.

Thank you for your comment. Prophylaxis with monoclonal antibodies was helpful in this context, leading to less infection, symptomatic or not, less hospitalization, less requirement of intensive care and lower mortality (Bertrand al. Efficacy of anti-SARS-CoV-2 monoclonal antibody prophylaxis and vaccination on the Omicron variant of COVID-19 in kidney transplant recipients. Kidney Int. 2022;102(2):440-442). In this era of immune-evasion capabilities, the continued evolution of omicron variants reinforces the need for new therapeutic monoclonal antibodies for COVID-19.

- All patients who developed COVID-19 survived, were there complications resulting from this?

Thank you for your comment. None of these patients developed complication related to covdi-19. None of these patients developed long COVID.

Reviewer 2 Report

Comments and Suggestions for Authors

The work by Bertrand presents an ELISPOT as a possible diagnostic tool in a certainly interesting clinical context. However, precisely for the proposed use, they should have at least proposed or mentioned in the discussions the comparison with methods based on the quantification of IFN already approved for other pathologies (IGRA tests). Moreover, Figure 1 should report appropriate statistics, otherwise, it is difficult to demonstrate the reliability of the method and to speculate on the data obtained from the samples of treated patients.

Other points to be addressed:

- Line 24: Why do the authors cite a cohort of 412 KTR, if their analysis was carried out only on a quarter of the samples? Please make this aspect explicit, or leave it only in the materials and methods.

- Lines 45-47: Please update the introduction also with FDA's guidelines on the use of the approved monoclonal antibodies against the Omicron variant.

- Figure 3: Statistics should be added.

- Figure 4: Please change the comma to the period in the statistical significance numbers reported in the graphs.

- Line 84: Typo "105", please correct it with "105".

- Line 156: Typo "with of the time", please correct it.

- Line 157: Typo "?", please correct it.

Author Response

The work by Bertrand presents an ELISPOT as a possible diagnostic tool in a certainly interesting clinical context.

- However, precisely for the proposed use, they should have at least proposed or mentioned in the discussions the comparison with methods based on the quantification of IFN already approved for other pathologies (IGRA tests).

Thank you for your comment. We now add in the discussion section the comparison with methods based on the quantification of IFN already approved for other pathologies (IGRA tests). 

“Assessment of T cell immunity, by ELISPOT or other Interferon-γ Release Assays for T-Cell-Mediated Immunity (IGRA) that can be easily implemented in a routine setting. At present, two such tests based on interferon-gamma release are available-Quan-T-Cell SARS-CoV-2 by EUROIMMUN and T-SPOT.COVID by Oxford Immunotec: the first head-to-head comparison of these two tests evaluating T-cell-mediated immunity against SARS-CoV-2 was recently published and show comparable results between the 2 tests”

- Moreover, Figure 1 should report appropriate statistics, otherwise, it is difficult to demonstrate the reliability of the method and to speculate on the data obtained from the samples of treated patients.

Thank you for your comment, we added appropriate statistics in Figure 1. Median number of reactive T cells (S1, S2, N, M and NS3A) was significatively higher in KTR with a history of SARS-CoV-2 infection compared to COVID-19 negative KTR (Figure 1a). Median number of spike-reactive T cells (S1+S2) was significatively higher in KTR with a history of symptomatic SARS-CoV-2 infection compared to asymptomatic infection (Figure 1b): 590 (IQR: 247.5-925) vs 145 (IQR: 60-507.5) SFC/106 CD3+ T cells (p=0.01). Median number of N-reactive T cells was not significatively higher: 245 (IQR: 100-452.5) vs 115 (IQR: 40-272.5) SFC/106 CD3+ T cells (p=0.11).

Other points to be addressed:

- Line 24: Why do the authors cite a cohort of 412 KTR, if their analysis was carried out only on a quarter of the samples? Please make this aspect explicit, or leave it only in the materials and methods.

We modified this point in the revised manuscript. We only leave the entire cohort of 412 KTR in the materials and methods section.

- Lines 45-47: Please update the introduction also with FDA's guidelines on the use of the approved monoclonal antibodies against the Omicron variant.

Thank you for your comment. We add this point in the introduction section:

“During the Covid-19 pandemic, the FDA granted EUAs for monoclonal antibodies thera-pies such as tixagevimab /cilgavimab and sotrovimab. However, both therapies lost their authorization due to their inefficacy against newer Omicron subvariants.”

- Figure 3: Statistics should be added.

Thank you for your comment, we added appropriate statistics in Figure 3.

- Figure 4: Please change the comma to the period in the statistical significance numbers reported in the graphs.

Thank you for your comment, we modified Figure 4 in the revised manuscript.

- Line 84: Typo "105", please correct it with "105".

We modified this point in the revised manuscript

- Line 156: Typo "with of the time", please correct it.

We modified this point in the revised manuscript

- Line 157: Typo "?", please correct it.

We modified this point in the revised manuscript